# THEval. Evaluation Framework for Talking Head Video Generation

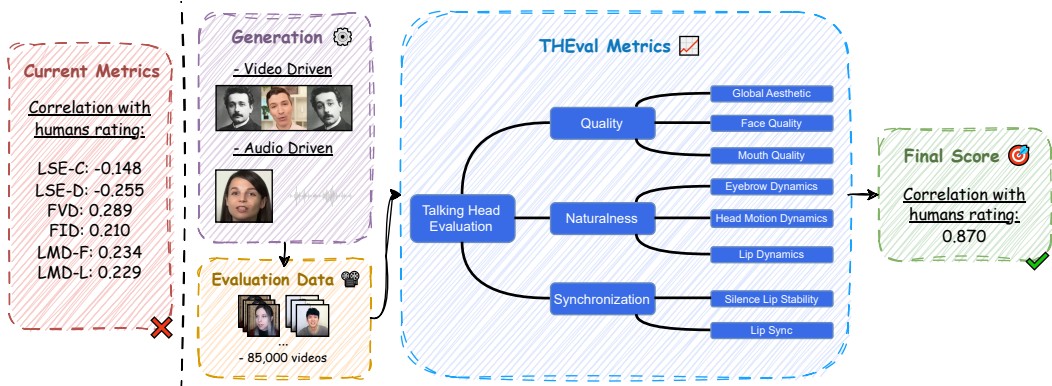

Figure 1: **Overview of the THEval benchmark.** We have generated talking head videos by 17 both, state-of-the-art video- and audio-driven methods, based on a dataset of over 5,000 videos spanning, resulting in 85,000 videos. We conduct a user study which demonstrates poor alignment between existing metrics (left red box) and human ratings. Motivated by this, we proceed to introduce the evaluation framework THEval, including 8 metrics related to (i) *quality*, (ii) *naturalness*, and (iii) *synchronization* (center). These metrics are combined into a final score (right green box) that showcases a high correlation of 0.870 with human ratings, thereby constituting a new benchmark for evaluation of talking head videos.

## ABSTRACT

Video generation has achieved remarkable progress, with generated videos increasingly resembling real ones. However, the rapid advance in generation has outpaced the development of adequate evaluation metrics. Currently, the assessment of talking head generation primarily relies on limited metrics, evaluating general video quality, lip synchronization, and on conducting user studies. Motivated by this, we propose a *new evaluation framework* comprising *8 metrics* related to three dimensions (i) *quality*, (ii) *naturalness*, and (iii) *synchronization*. In selecting the metrics, we place emphasis on efficiency, as well as alignment with human preferences. Based on this considerations, we streamline to analyze fine-grained dynamics of head, mouth, and eyebrows, as well as face quality. Our extensive experiments on 85,000 videos generated by 17 state-of-the-art models suggest that while many algorithms excel in lip synchronization, they face challenges with generating expressiveness and artifact-free details. These videos were generated based on a novel real dataset, that we have curated, in order to mitigate bias of training data. Our proposed benchmark framework is aimed at *evaluating the improvement of generative methods*. Original code, dataset and leaderboards will be publicly released and regularly updated with new methods, in order to reflect progress in the field.

# 1 INTRODUCTION

Generative models have witnessed remarkable progress and are currently able to generate high-resolution, highly realistic images (Xu et al., 2024b; Xue et al., 2024) and videos (Wang et al., 2024b;a). However, the rapid advancement in generation has outpaced the development of adequate evaluation metrics, see details on existing metrics and their limitations in Appendix F. Prominent metrics in image quality evaluation, such as the Fréchet Inception Distance (FID), Inception Score (IS), and Learned Perceptual Image Patch Similarity (LPIPS) are limited in capturing quality and realism in generated data. Despite their limitations, most recent works in *image* generation are predominantly evaluated by the means of *FID and IS* (Xu et al., 2024b; Yang et al., 2024), as well as *human user studies* (Shi et al., 2024; Chen et al., 2024b).

**Video quality evaluation** encounters similar challenges as image quality assessment, primarily relying on metrics such as FID, Fréchet Video Distance (FVD), and Inception Score (IS) (Wang et al., 2024a; Qing et al., 2024), which have limitations for high-quality videos. These metrics often omit motion quality and temporal coherence, necessitating human evaluation for such aspects. Alternative metrics (Liu et al., 2024; Huang et al., 2024) have been proposed for *general* video quality evaluation, assessing imaging and aesthetic quality, as well as for background consistency and motion smoothness in the context of temporal evaluation.

Deviating from general text-based video generation, *talking head (TH) generation* involves an audio-speech sequence, employed to animate a face image. Evaluating this specific setting requires assessment of global image quality, as well as the quality of facial motions, including lip synchronization, facial expression, and head pose movement, all contributing to the video's overall naturalness. Currently, evaluations (Xu et al., 2024a; Chen et al., 2024c) focus on two main parameters: *image quality*, measured by metrics including FID, FVD, SSIM, and PSNR, which share issues found in other video generation tasks, as well as *lip synchronization quality*, assessed using Euclidean distances between landmarks or scores from the pretrained Syncnet neural network (Chung & Zisserman, 2016). However, studies have shown that Syncnet is unstable (Yaman et al., 2024b) and sensitive to factors such as mouth cropping and head pose. Additionally, the commonly used Syncnet confidence score (LSE-C) and distance (LSE-D) have been found to correlate poorly with human preferences (Zhang et al., 2024).

Motivated by the above and towards addressing the challenges of evaluating TH videos, we propose a new framework, referred to as THEVAL. Our main contributions include the following.

- We introduce THEVAL, a new framework with 8 fine-grained metrics across three dimensions (i) quality, (ii) naturalness, and (iii) synchronization that shows a Spearman correlation coefficient of $\rho = 0.870$ between our *Final Score* and human ratings.
- We release a new, challenging evaluation dataset of over 5000 videos designed to test model generalization on unseen videos.
- Through an extensive benchmark of 17 SOTA audio- and video-driven models, and provide a detailed analysis of their strength and weaknesses.
- We conduct a user study, where participants compare (a) and (b), showcasing that THEval strongly correlates with human preferences. It also reveals that all current metrics do not well correlate with user preferences.

Ultimately, we here make the case that existing metrics for evaluating talking head videos are highly limited and proceed to propose a novel evaluation framework THEval (see Figure 1), which aligns with human preferences. THEval is intended for researchers and practitioners in the field of talking head generation, providing a tool that identifies remaining challenges and fosters progress. Towards this, We will make our benchmark publicly available, which includes dataset, code, and a live leaderboard.

# 2 RELATED WORK

## 2.1 TALKING HEAD VIDEO GENERATION

TH video generation can be *video* or *audio*-driven. Video-driven methods animate a face image using a driving video, effectively replacing the identity in the original footage. Early techniques

utilized generative adversarial networks (GAN)-inversion or motion flow, while recent approaches focus on directly controlling the latent space (Ni et al., 2023; Wang et al., 2024c). Such methods tend to outperform audio-driven methods, as they reconstruct the motion and therefore encompass fewer degrees of freedom. In contrast, audio-driven methods entail fewer constraints in their effort to reproduce lip motion, expression, and head pose corresponding to the audio input. *Text*-driven models (Wang et al., 2022a; Li et al., 2021) commonly convert text to audio or phonemes before generating video, and therefore fall in this context under the category of audio-driven methods.

In this paper, we focus on audio- and video-driven TH generation that animate face images in RGB space (Zhou et al., 2021; Wang et al., 2023), employing landmarks (Suzhen et al., 2021; Gururani et al., 2023), or face mesh representations (Zhang et al., 2021; Thies et al., 2020). While some methods produce 3D meshes, we note that THEval focuses solely on RGB videos. Audio-driven methods can be based on CNNs, LSTMs, or more recent diffusion models (Yu et al., 2023a;b; Shen et al., 2023). Challenges in generation of TH have to do with accurate lip synchronization, as well as realistic face appearance, expressions and head movements. While state-of-the-art methods improve in generating related videos, evaluating such aspects remains an open challenge.

## 2.2 EVALUATION OF TALKING HEAD VIDEOS

**Image quality based metrics.** Currently image quality is evaluated with metrics such as FID, SSIM, PSNR (Xu et al., 2024a; Chen et al., 2024c). FID compares the probability density distributions of real and generated images, SSIM measures the similarity between real and generated images, and PSNR evaluates noise levels in the output videos. Such metrics face challenges with complex generated videos and are impacted by factors that do not affect video quality. Moreover, these metrics often do not align with human ratings for high-quality images and videos. In the context of audio-driven TH generation, experiments typically use small subsets of videos (a few hundred to a few thousand) in inference for efficiency. However, metrics such as FID (and by extension FVD) can be biased when assessed on limited samples, which may not provide a sufficient basis for generalization.

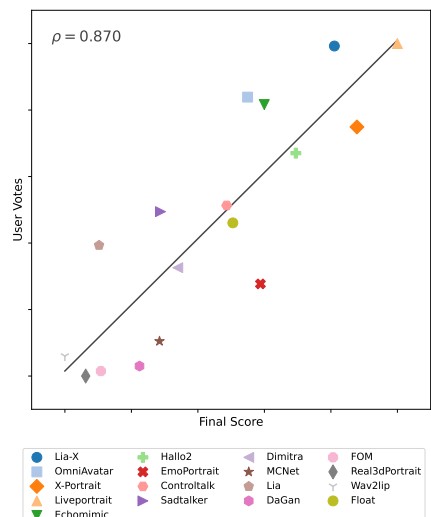

**Facial landmark based metrics.** The LMD-F and LMD-M metrics assess facial expression and head movement using facial landmarks, showing better correlation with human evaluations than other metrics (Zhang et al., 2024). However, they impart significant limitations. LMD-M penalizes small temporal lags that human evaluators might not notice, whereas LMD-F imposes strong penalties for differences in head motion and expression between generated videos and ground truth. This penalization is not justified as head motion and facial expression exhibit only weak correlation with audio sequences. We note that naturalness yields high human ratings, even in the presence of such discrepancies. Nevertheless, LMD metrics remain widely adopted, and the LMD-F variant is sometimes employed in evaluations involving real head poses, where it often produces favorable outcomes (Ma et al., 2023a;b).

Figure 2: **THEval–Human Correlation.** A high Spearman correlation coefficient ($\rho = 0.870$) confirms THEval's strong alignment with human ratings. Each point represents a human preference for a state-of-the-art model win rate (y-axis) versus its THEval score (x-axis). This validation enables THEval to serve as an efficient proxy for costly user studies.

**Syncnet metrics.** The most prominent metrics for lip synchronization are the Syncnet distance (LSE-D) and confidence score (LSE-C). Syncnet, a CNN-based network, aims to capture the correlation between audio and spatio-temporal features of the mouth region, calculating the audio offset (the number of frames by which audio is early or late compared to video). LSE-D represents the feature distance at the predicted offset, whereas LSE-C measures the difference between the minimum and

median distances across various offsets. Syncnet metrics are effective for determining *audio offsets* and identifying speakers in multi-person videos. Limitations of LSE-C and LSE-D include (1) highly limited correlation with human evaluation (Zhang et al., 2024), (2) unreasonable values, *e.g.,* ground truth was widely outperformed by generation results (Xu et al., 2024a), (3) instability to factors such as the cropped mouth region, image quality and brightness (Yaman et al., 2024b;a).

**User studies.** Given the above limitations of objective metrics, user studies remain a viable evaluation for TH generation. However, associated limitations include the tedious and time-consuming nature of such studies.

Current methods often use a few limited metrics, to summarize the complex process of generating TH. Relying on single metrics can render aspects related to over or underperformance not interpretable. Such aspects can be of different nature, *e.g.,* accurate mouth movements, realistic head motion, as well as overall appearance.

Motivated by the above gap, we propose THEVAL, a framework developed to enhance existing metrics for evaluating TH videos. Rather than relying on a limited set of general metrics for assessing performance of TH-generation models, we advocate for a detailed breakdown of relevant factors, encompassing global head movement to nuanced expression. This decomposition facilitates a more comprehensive understanding, enabling targeted improvements in future models.

## 3 METRICS IN THEVAL EVALUATION FRAMEWORK

We introduce our proposed THEVAL, which combines 8 algorithmic and perceptual metrics. Drawing from recent video generation benchmarks (Huang et al., 2024), our framework entails 3 core dimensions representing (i) *quality*, (ii) *naturalness*, as well as (iii) *synchronization*. A visual representation of the results is available in Figure 3 and an in-depth formulation is described in Appendix B. We proceed with the motivation and implementation of each metric.

### 3.1 QUALITY

This dimension reflects the perceptual appeal of a generated TH, with emphasis on clarity, sharpness, and color fidelity.

**(1) Global Aesthetics.** We introduce a global aesthetics measure by adopting the Image Aesthetic Assessment (IAA) component of TOPIQ (Chen et al., 2024a), which accounts for high-level attributes such as composition, lighting, and color harmony, with $S_{aes,j}$ denoting the aesthetic score and $N$ the total number of frames

$$Global\ Aesthetics = \frac{1}{N} \sum_{j=1}^{N} S_{aes,j}. \tag{1}$$

**(2), (3) Mouth- and Face-Centric Quality.** We propose a region-specific assessment strategy. Specifically, we compute Image Quality Assessment (IQA) with TOPIQ for the full face and MUSIQ (Ke et al., 2021) for the mouth region. To this end, we localize facial regions via landmarks and extract IQA scores separately, with $Q$ denoting the quality score for the face and mouth

$$Face\ /\ Mouth\ Quality = \frac{1}{N} \sum_{j=1}^{N} Q_{face/mouth,j}. \tag{2}$$

Separating the mouth from the face allows us to explicitly quantify the distinct difficulty of synthesizing realistic mouth motion in generative video models.

### 3.2 NATURALNESS

Naturalness indicates the realism of facial behavior in TH videos. Prior work (Hauser et al., 2024) shows that subtle facial asymmetries such as in eyebrow, mouth, or head motion enhance perceived believability, appeal, and naturalness. Motivated by these findings, we include metrics that quantify

lip, eyebrow, and head movement dynamics to assess whether TH videos exhibit realistic and engaging facial behavior.

**(4)** The **Lip Dynamics** metric captures how natural and varied mouth movements appear over time. For each frame, we detect the face and extract 40 lip landmarks. From these points, we compute pairwise distances that describe the lip shape and configuration. By tracking how these distances change across frames, the metric quantifies the variability and dynamics of lip motion throughout the video.

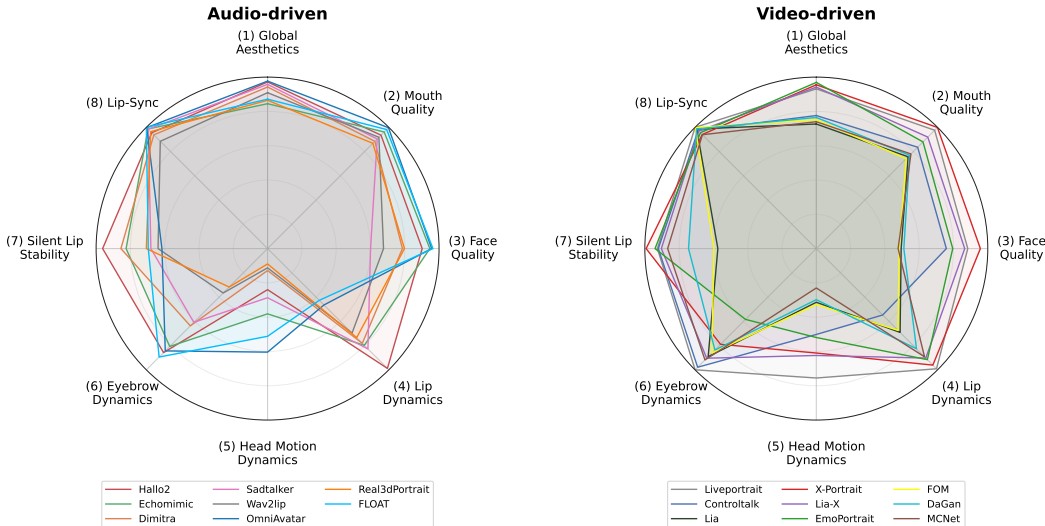

Figure 3: **Quantitative comparison of audio-driven (left) and video-driven (right) models on the THEval framework.** The radar charts visualize performance across our eight evaluation metrics, revealing distinct performance profiles. Video-driven models generally achieve more balanced, high-quality results, while audio-driven models exhibit greater variance, often excelling in dynamics but struggling with overall naturalness. Scores that are farther from the center indicate superior performance.

**(5)** The **Head Motion Dynamics** metric quantifies head movements in a video. We first estimate the head's orientation, specifically, its pitch, yaw, and roll, as well as the position of the head within the frame. For each video segment, the metric follows changes in head pose angles and translations over time. It is defined as follows, where $\overline{\sigma_{\text{angle}}}$ is the mean standard deviation of pitch, yaw, and roll angles, $\overline{V_{\Delta\text{angle}}}$ is the mean variance of their first-order temporal differences, and $\overline{V_{\text{trans}}}$ is the mean variance of face center translations

$$Head\ Motion\ Dynamics = \sqrt{\left(\overline{\sigma_{\text{angle}}} \cdot \overline{V_{\Delta\text{angle}}}\right) + \overline{V_{\text{trans}}}}. \tag{3}$$

**(6)** The **Eyebrow Dynamics** metric captures the variability of eyebrow movements in a video. For each frame, we detect facial landmarks and calculate the relative distance between eyebrows and eyes, normalized by the distance between eyes. The metric captures raising or lowering eyebrows that are pertinent in conveying expressions and emotions. This metric reflects the average intensity of these micro-movements throughout the video.

## 3.3 SYNCHRONIZATION

Synchronization assesses the extent to which a TH's lip movements correspond with the accompanying speech. Previous research (Chae-Yeon et al., 2025) indicates that viewers are sensitive to misalignment and tend to favor scenarios where the intensity of lip and jaw movements corresponds to the audio volume. Building on these findings, we incorporate metrics that evaluate lip stability during silent intervals and the alignment of mouth openness with speech intensity to determine whether TH videos demonstrate realistic and expressive lip synchronization.

**(7)** The **Silent Lip Stability** relates to the mouth movement during silent periods. Silent frames are detected using a VAD model, considering only segments longer than 300 ms. For each frame, facial landmarks are extracted, and the vertical distances between upper and lower lips are computed and normalized by the interocular distance. Stability is quantified using the median absolute deviation of these distances across all silent frames.

**(8)** The **Lip-Sync** metric evaluates the degree of alignment between mouth movements and the spoken audio. First, we use the same VAD model as *(7) Silent Lip Stability* to identify frames containing speech. For each frame, we compute the mouth openness $O_t$ by measuring the normalized distance between the upper and lower lip landmarks, and the corresponding audio volume $V_t$ from the Root-Mean-Square (RMS) energy. Both the mouth openness and audio volume signals are independently normalized to a [0, 1] range. The final lip-sync metric is the mean absolute difference between these two normalized signals. The lip-sync metric $L_{sync}$ is calculated as

$$L_{sync} = \frac{1}{S} \sum_{t \in S} \left| \frac{O_t - \min(O_S)}{\max(O_S) - \min(O_S)} - \frac{V_t - \min(V_S)}{\max(V_S) - \min(V_S)} \right|, \tag{4}$$

where $S$ is the set of speech frames, and $O_S$ and $V_S$ are the sets of mouth openness and volume values for those frames, respectively.

**Final Score** To provide a single, interpretable measure for evaluating TH-videos, we compute a *final composite score* that aggregates performance across the three key dimensions (i) *quality*, (ii) *Naturalness*, and (iii) *Synchronization*. This score is derived from the eight metrics that we introduced, which are first normalized relative to a ground-truth (GT) reference and then averaged into a single value. For normalization, a model's score on a specific metric is determined by its similarity to the ground truth (GT) value, as expressed by the following equation.

$$s = 1 - \frac{|\text{Model Score} - \text{GT Score}|}{\text{GT Score}}, \tag{5}$$

where a score of 1 indicates a perfect match with the ground truth, lower values reflect increasing deviation.

## 4 EXPERIMENTS

We present a series of experiments designed to validate the effectiveness of our proposed evaluation framework, including the new evaluation dataset, the methods compared, and the correlation with human ratings.

### 4.1 THEVAL DATASET

To thoroughly assess the generalization capabilities of contemporary TH models, we present the THEVAL DATASET, a new benchmark designed to highlight the benefits and limitations of the models under evaluation. Our primary goal was to create an evaluation set with samples explicitly not seen during the training of the models we evaluate. The dataset was constructed by sourcing 5,011 video from a wide range of 31 public YouTube channels, ensuring a rich variety of content across multiple languages, including Spanish, Italian, English, French, Japanese, and Chinese. Each video have a single speaker, a clear and primarily frontal view of the face, and high-fidelity 1080p resolution. This resulted in a final dataset of over 18 hours of content, with an average video length of 13 seconds. Visual examples showcasing this diversity are presented in Figure 6.

### 4.2 SETUP

**Compared Methods.** We evaluate the following 17 state-of-the-art TH generation models for video-driven approaches, **Controltalk** (Zhao et al., 2024), **Liveportrait** (Guo et al., 2024), **LIA** (Wang et al., 2022b), **X-Portrait** (Xie et al., 2024), **LIA-X** (Wang et al., 2025), **EmoPortrait** (Drobyshev et al., 2024), **MCNet** (Hong & Xu, 2023), **DaGan** (Hong et al., 2022), **FOM** (Siarohin et al., 2019) and audio-driven approaches **Hallo2** (Cui et al., 2024), **EchoMimic** (Chen et al., 2024c), **Wav2Lip** (Prajwal et al., 2020), **SadTalker** (Zhang et al., 2023), **Dimitra** (Chopin et al., 2025), **OmniAvatar** (Gan et al., 2025), **Real3dPortrait** (Ye et al., 2024) and **FLOAT** (Ki et al., 2024).

Figure 4: **Visual examples from our new THEval dataset.** Our benchmark is curated for diversity, featuring a wide range of subjects, head poses, and expressions from multiple linguistic backgrounds (including Spanish, Italian, English, French, Japanese, and Chinese). This dataset is specifically designed to test the generalization capabilities of talking head generation models on truly unseen data.

Details associated to the above methods are provided in Appendix E. Each method is executed with its default hyperparameters, and model weights provided by the authors or official repositories. For a fair comparison, we provide the same audio and reference frames for each method, in order to generate videos. We do not evaluate 3D Gaussian Splatting or Neural Radiance Fields methods, as they require multi-view inputs and are limited to a fixed set of pretrained identities, making them unsuitable for our single-view, multi-identity setting.

Table 1: **Evaluation results of 17 TH Generation Models on the THEval Dataset.** We compare state-of-the-art audio-driven and video-driven models across our proposed categories of (i) *quality*, (ii) *synchronization*, and (iii) *naturalness*. The results highlight the distinct performance profiles of the two approaches. For each metric, higher values indicate better performance. The scores in bold represent the three best scores per dimension.

| | Model | Quality | | | Synchronization | |
|---|---|---|---|---|---|---|
| | | (1) Global Aesthetics ↑ | (2) Mouth Quality ↑ | (3) Face Quality ↑ | (7) Silent Lip Stability ↑ | (8) Lip-Sync ↑ |
| **Audio Driven** | Hallo2 (Cui et al., 2024) | **0.9619** | 0.9254 | 0.9017 | **0.9620** | 0.9502 |
| | OmniAvatar (Gan et al., 2025) | **0.9767** | **0.9919** | **0.9521** | 0.6160 | **0.9972** |
| | Echomimic (Chen et al., 2024c) | 0.8499 | **0.9617** | **0.9514** | **0.8251** | **0.9964** |
| | FLOAT (Ki et al., 2024) | 0.8713 | **0.9868** | **0.9645** | 0.6958 | **0.9992** |
| | Sadtalker (Zhang et al., 2023) | **0.9576** | 0.9142 | 0.6005 | 0.6806 | 0.9794 |
| | Dimitra (Chopin et al., 2025) | 0.9523 | 0.8798 | 0.7914 | **0.8555** | 0.9430 |
| | Real3dPortrait (Ye et al., 2024) | 0.8597 | 0.8732 | 0.7934 | 0.7072 | 0.9695 |
| | Wav2lip (Prajwal et al., 2020) | 0.9090 | 0.9180 | 0.6762 | 0.6388 | 0.8849 |
| **Video Driven** | LIA-X (Wang et al., 2025) | **0.9466** | 0.9195 | 0.8705 | 0.9087 | 0.9644 |
| | Liveportrait (Guo et al., 2024) | 0.9464 | **0.9760** | **0.8784** | **0.9316** | **0.9980** |
| | X-Portrait (Xie et al., 2024) | **0.9502** | **0.9990** | **0.9568** | **0.9924** | 0.9407 |
| | EmoPortrait (Drobyshev et al., 2024) | **0.9542** | 0.8799 | 0.7957 | **0.9354** | 0.9608 |
| | Controltalk (Zhao et al., 2024) | 0.7759 | 0.8360 | 0.7584 | 0.9163 | 0.9897 |
| | MCNet (Hong & Xu, 2023) | 0.7499 | 0.7655 | 0.4771 | 0.8669 | 0.9541 |
| | DaGan (Hong et al., 2022) | 0.7547 | 0.7646 | 0.5105 | 0.7452 | 0.9719 |
| | LIA (Wang et al., 2022b) | 0.7265 | 0.7622 | 0.4899 | 0.5741 | **0.9913** |
| | FOM (Siarohin et al., 2019) | 0.7516 | 0.7566 | 0.4875 | 0.5970 | **0.9929** |

| | Model | Naturalness | | | Final Score ↑ |
|---|---|---|---|---|---|
| | | (4) Lip Dynamics ↑ | (5) Head Motion ↑ | (6) Eyebrow Dynamics ↑ | |
| **Audio Driven** | Hallo2 (Cui et al., 2024) | **0.9883** | 0.2395 | **0.8530** | **0.8477** |
| | OmniAvatar (Gan et al., 2025) | 0.4650 | **0.6039** | 0.8488 | **0.8064** |
| | Echomimic (Chen et al., 2024c) | **0.7930** | **0.3806** | 0.8071 | **0.8207** |
| | FLOAT (Ki et al., 2024) | 0.4266 | **0.5115** | **0.8945** | 0.7938 |
| | Sadtalker (Zhang et al., 2023) | **0.8276** | 0.2867 | 0.6084 | 0.7319 |
| | Dimitra (Chopin et al., 2025) | 0.7863 | 0.1279 | 0.6372 | 0.7467 |
| | Real3dPortrait (Ye et al., 2024) | 0.7348 | 0.0895 | 0.3170 | 0.6680 |
| | Wav2lip (Prajwal et al., 2020) | 0.6966 | 0.1124 | 0.3662 | 0.6502 |
| **Video Driven** | LIA-X (Wang et al., 2025) | 0.9030 | **0.6233** | 0.9090 | **0.8806** |
| | Liveportrait (Guo et al., 2024) | **0.9913** | **0.7548** | **0.9997** | **0.9345** |
| | X-Portrait (Xie et al., 2024) | **0.9611** | **0.6091** | 0.7897 | **0.8999** |
| | EmoPortrait (Drobyshev et al., 2024) | **0.9159** | 0.5136 | 0.5840 | 0.8174 |
| | Controltalk (Zhao et al., 2024) | 0.5476 | 0.5058 | **0.9785** | 0.7885 |
| | MCNet (Hong & Xu, 2023) | 0.8925 | 0.2297 | **0.9132** | 0.7311 |
| | DaGan (Hong et al., 2022) | 0.8262 | 0.3029 | 0.8362 | 0.7140 |
| | LIA (Wang et al., 2022b) | 0.6912 | 0.3080 | 0.8920 | 0.6794 |
| | FOM (Siarohin et al., 2019) | 0.6743 | 0.3269 | 0.8613 | 0.6810 |

**Implementation Details.**   We employ MediaPipe Face Mesh (Lugaresi et al., 2019) for extracting facial landmarks, including eyes, lips, and eyebrows, across all frames. Head pose metrics are computed using FaceXFormer (Narayan et al., 2024). Finally, to detect speech segments in the audio we use Silero VAD (Silero, 2024).

**Ground Truth as Reference.**   We include comparisons with ground-truth GT videos as reference, in order to compute absolute differences with the generation methods. This allows us to quantitatively evaluate the resemblance of each method to real facial behavior. For instance, in metrics such as (5) Head Motion Dynamics, a low score indicates that the generated head movements are too subtle as opposed to ground truth, whereas a high score suggests exaggerated head motions.

## 4.3 THEval Correlation with Human Rating

Towards validating our metrics, we conduct a user study in Hugging Face Space. Participants are asked to evaluate paired real and generated videos based on the same audio. Pairing included all combinations, *i.e.,* real and generated videos, as well as generated videos pertained to all combinations of the seventeen state-of-the-art methods. Participants were instructed by *"Please watch both videos and select which one looks more realistic"*. The instruction aims at minimizing cognitive load and reducing rating errors, ensuring participants could make consistent and intuitive judgments. In total, we acquired 3,519 ratings, distributed equally among the seventeen models. We then compute the Spearman correlation coefficient between normalized metric scores and human preference scores. We observe strong alignment between our THEval metrics and human opinion scores, as reported in Figure 2 and Table 2. The *final THEval score* is highly correlated with human ratings, with a strong correlation of $\rho = 0.870$. In addition, the individual metrics exhibit high correlations with human preferences. We note that (4) Lip Dynamics and (8) Lip-Sync are of lower correlation, which is expected as human perception of realism integrates multiple cues reflected by the 8 metrics. By combining complementary *expert* metrics, our framework achieves a strong aggregated alignment with human preference, advocating for the composite design of THEval.

Table 2: **Correlation between metrics and human ratings.** We report Spearman's rank correlation coefficient ($\rho$), p-values, and 95% Confidence Intervals (CI) between each metric and human preferences. The 95% CIs are obtained via bootstrapping with $n = 10,000$ resamples. The results clearly show that our proposed metrics and final scores have a strong, significant alignment with human ratings.

| Metric | Correlation ($\rho$) | p-value | 95% CI |
|---|---|---|---|
| LSE-C | -0.164 | 0.530 | [-0.613, 0.388] |
| LSE-D | -0.269 | 0.297 | [-0.675, 0.282] |
| FVD | 0.289 | 0.260 | [-0.321, 0.782] |
| FID | 0.210 | 0.416 | [-0.344, 0.710] |
| LMD-F | 0.231 | 0.389 | [-0.392, 0.775] |
| LMD-L | 0.227 | 0.399 | [-0.389, 0.759] |
| (1) Global Aesthetics | 0.544 | 0.020 | [0.129, 0.795] |
| (2) Mouth Visual Quality | 0.765 | < 0.001 | [0.498, 0.917] |
| (3) Face Quality | 0.699 | 0.001 | [0.430, 0.875] |
| (4) Lip Dynamics | 0.414 | 0.088 | [-0.155, 0.769] |
| (5) Head Motion Dynamics | 0.763 | < 0.001 | [0.418, 0.942] |
| (6) Eyebrow Dynamics | 0.527 | 0.025 | [0.060, 0.856] |
| (7) Silent Lip Stability | 0.484 | 0.042 | [0.033, 0.808] |
| (8) Lip-Sync | 0.404 | 0.097 | [-0.143, 0.775] |
| Quality | 0.713 | 0.001 | [0.424, 0.895] |
| Naturalness | 0.702 | 0.001 | [0.217, 0.862] |
| Synchronization | 0.603 | 0.008 | [0.323, 0.919] |
| Final Score | 0.870 | < 0.0001 | [0.648, 0.967] |

In contrast, SyncNet, landmark-based metrics, FID, and FVD demonstrate minor to no alignment with human ratings. These results indicate that our approach provides a reliable alternative to current evaluation metrics.

Further details on the user study including the user study interface and algorithm details to ensure fair evaluation are in Appendix A.

## 5 DISCUSSION

### 5.1 AUDIO-DRIVEN METHODS

We note that audio-driven approaches consistently struggle *w.r.t.* facial expressiveness, as well as head pose movement, which is reflected in the videos. In addition, FLOAT and OmniAvatar tend to exaggerate mouth movements. In the case of OmniAvatar, this is due to the employed image-to-video model WanVideo (Wan et al., 2025), which generates exaggerated articulations that appear unnatural, reflected in the scores of metrics (4) Lip Dynamics and (7) Silent Lip Stability. Furthermore, we observe that in longer videos, OmniAvatar exhibits temporal drift, with the facial identity gradually diverging from the source, visual artifacts becoming increasingly visible, and a perceptible orange color emerge, all of which reduce overall video quality. This temporal instability prevents OmniAvatar from achieving the highest ranking in our benchmark, despite its otherwise strong performance. Visual examples are availabe in Appendix D.

Naturally, newer audio-driven methods perform well *w.r.t.* overall video quality and visual appeal compared to earlier approaches, thanks to advances in backbone architectures and training data. However, amplified expressions pose a remaining challenge. By disentangling synchronization, expressiveness, and quality, THEval effectively highlights these nuanced behavior.

### 5.2 VIDEO-DRIVEN METHODS

Video-driven methods, by contrast to audio-driven ones, exhibit stronger expressivity, while also maintaining reliable synchronization. This is due to the fact that motion priors from driving videos allow for generation of realistic head dynamics and subtle facial expressions. We observe that earlier video-driven models incorporate visible artifacts, particularly in case of large head movements. Such artifacts manifest as tearing, blur, or instability in the facial regions, leading to lower overall quality scores in our benchmark. This trade-off between expressivity and quality is effectively highlighted by our proposed metrics, which separate naturalness-related measures from face and mouth quality assessments.

Overall, the best-performing video-driven models showcase a strong balance between expressivity, synchronization, and visual fidelity, obtaining a higher *Final Score* in THEval. Our results indicate that the *Final Score* reflects these trade-offs well, ranking methods based on human perception.

## 6 CONCLUSIONS

We introduced a comprehensive benchmark and an evaluation dataset, referred to as THEVAL, streamlined to evaluate generated talking head (TH) videos. The eight metrics in THEval cover the three key dimensions (i) quality, (ii) naturalness, and (iii) synchronization. Our benchmark enables both, fine-grained and efficient assessment of TH videos. Experiments show that THEVAL metrics correlate strongly with human preference, unlike currently used metrics, which often fail to reflect perceptual ratings. We further observe that state-of-the-art audio- and video-driven generative models still face challenges in producing realistic lip movements, natural expressiveness, and artifact-free rendering. Specifically, audio-driven methods have advanced *w.r.t.* synchronization, however often lack natural head motion and may incorporate exaggerated expressions. At the same time recent video-driven counterparts generate more expressive and realistic videos. Our THEval metrics capture this illustratively, and by aggregating them into a *Final Score*, THEval provides both, detailed diagnostics and a final measure that matches human preference, rendering it a much-needed benchmark for talking head generation. We aim at fostering the development of new generation methods. Future work will extend the benchmark to more diverse scenarios, such as multiple humans and side views.

# 7 ETHICS STATEMENT

This work introduces THEval, a benchmark for talking head generation, with careful consideration of its ethical implications. The dataset was sourced from public videos and curated for diversity to ensure fair evaluation. An anonymized user study was conducted to validate the metrics. We acknowledge the potential for misuse of "deepfake" technology and aim to contribute to its responsible development. By providing a robust, human-aligned evaluation framework, THEval helps researchers better understand model capabilities and limitations which is a step for creating effective detection methods and safeguards. To foster transparency, we will release the full benchmark, dataset, and code.

# 8 REPRODUCIBILITY STATEMENT

To ensure the reproducibility of our work, we will make our code, the THEval dataset, and a regularly updated leaderboard publicly available. Section 4 of the main paper provides a comprehensive description of our experimental setup, including the dataset creation process (Section 4.1), the models evaluated, and implementation details (Section 4.2). Further mathematical details for all proposed metrics are available in Appendix B, and a thorough description of the user study can be found in Appendix A. The specific state-of-the-art methods used for benchmarking are detailed in Appendix E, and all models were used with their official, publicly available code and default hyperparameters.

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

# A APPENDIX: USER STUDY DETAILS

## A.1 WEBSITE AND ALGORITHM DETAILS

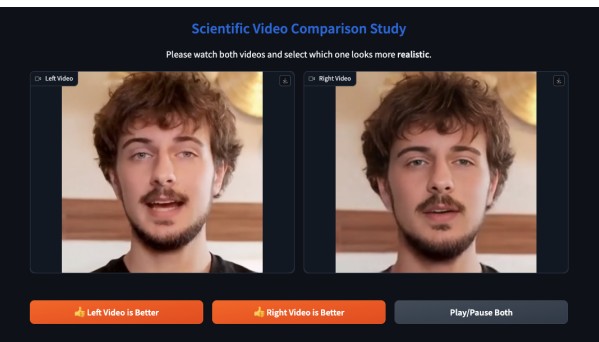

Figure 5: **Screenshot of our user study interface.** The interface was designed to be intuitive and easy to use for human raters. Both videos can be played simultaneously using the *Play/Pause Both* button, and participants indicate which video appears more realistic by selecting one of the two highlighted choice buttons.

To ensure a fair and unbiased comparison between methods, the algorithm selects videos randomly from common videos of all methods. By choosing videos at random, the evaluation avoids over-representing any specific content or scenario, which could otherwise skew results. Additionally, the algorithm randomly assigns each method to the left or right position in the user interface for each comparison to mitigates positional bias, ensuring that participants preferences are based on the video quality itself rather than the side on which it appears. Screenshot of the website is available on Figure 5.

# B APPENDIX: MATHEMATICAL DETAILS OF METRICS

This appendix provides more detailed mathematical explanations for the 8 metrics used in the THEval framework. These metrics are categorized into *video quality*, *naturalness*, and *synchronization*.

## B.1 VIDEO QUALITY METRICS

**(1) Global Aesthetics** Global Aesthetics are assessed using the Image Aesthetic Assessment (IAA) component of the TOPIQ model (Chen et al., 2024a), which is pre-trained on the AVA dataset. Let $I_j$ be the $j$-th frame of a video with $N$ frames. The TOPIQ IAA model, denoted $f_{IAA}$, computes an aesthetic score for each frame:

$$S_{aes,j} = f_{IAA}(I_j) \tag{6}$$

This score reflects properties such as composition, lighting, and color harmony. The final Global Aesthetics score for the video is the average of these per-frame scores:

$$Global\ Aesthetics = \frac{1}{N} \sum_{j=1}^{N} S_{aes,j} \tag{7}$$

**(2) Face Quality** Face Quality is evaluated using the Image Quality Assessment (IQA) component of the TOPIQ model (Chen et al., 2024a), pre-trained on the CGFIQA dataset. For each frame $I_j$, the face region $F_j$ is detected and cropped. The TOPIQ IQA model for faces, $f_{IQA_{face}}$, then computes a quality score:

$$Q_{face,j} = f_{IQA_{face}}(F_j) \tag{8}$$

The Face Quality score for the video is the average of these per-frame face quality scores:

$$Face\ Quality = \frac{1}{N} \sum_{j=1}^{N} Q_{face,j} \tag{9}$$

**(3) Mouth Quality** Mouth Quality is assessed using the MUSIQ (Multi-scale Image Quality Transformer) model (Ke et al., 2021), specifically its IQA variant pre-trained on the SPAQ dataset. For each frame $I_j$, the mouth region $M_j$ is detected using facial landmarks and subsequently cropped. The MUSIQ IQA model, $f_{IQA_{mouth}}$, predicts a quality score for this region:

$$Q_{mouth,j} = f_{IQA_{mouth}}(M_j) \tag{10}$$

The Mouth Quality score for the video is the average of these per-frame mouth region quality scores:

$$Mouth\ Quality = \frac{1}{N} \sum_{j=1}^{N} Q_{mouth,j} \tag{11}$$

### B.2 NATURALNESS METRICS

**(4) Lip Dynamics** This metric quantifies the variability of lip movements. For each frame $j$, $K$ lip landmarks $L_{j,k} = (x_{j,k}, y_{j,k})$ for $k = 1, \ldots, K$ (where $K = 40$) are extracted. A feature vector $s_j = (d_{j,1}, d_{j,2}, \ldots, d_{j,M})$ consisting of $M$ selected pairwise Euclidean distances between these landmarks is computed for each frame $j$. These distances capture the lip shape. The variability for each distance component $m$ across $N$ frames is its standard deviation:

$$\sigma_m = \sqrt{\frac{1}{N-1} \sum_{j=1}^{N} (d_{j,m} - \bar{d}_m)^2} \tag{12}$$

where $\bar{d}_m = \frac{1}{N} \sum_{j=1}^{N} d_{j,m}$ is the mean of the $m$-th distance component over all frames. The Lip Dynamics score is the average of these standard deviations, representing the overall variability in lip shape:

$$Lip\ Dynamics = \frac{1}{M} \sum_{m=1}^{M} \sigma_m \tag{13}$$

**(5) Head Motion Dynamics** The Head Motion Dynamics metric quantifies the complexity and dynamism of head movements by combining measures of angular motion range, variability of angular velocities, and variability of head position within the frame. For a video with $N$ frames, the process begins with estimating head pose parameters for each frame $j = 1, \ldots, N$. These parameters include head orientation angles: pitch ($\theta_{p,j}$), yaw ($\theta_{y,j}$), and roll ($\theta_{r,j}$), all expressed in degrees. Additionally, the 2D coordinates of the center of the detected face ($c_{x,j}, c_{y,j}$) in the frame are determined. Let $P = \theta_{p,1}, \ldots, \theta_{p,N}, Y = \theta_{y,1}, \ldots, \theta_{y,N}$, and $R = \theta_{r,1}, \ldots, \theta_{r,N}$ denote the sequences of pitch, yaw, and roll angles, respectively. Similarly, $T_x = c_{x,1}, \ldots, c_{x,N}$ and $T_y = c_{y,1}, \ldots, c_{y,N}$ represent the sequences of x and y coordinates of the face center.

To assess the range of angular motion, the standard deviation for each angular sequence is calculated:

$$\sigma_P = \text{std}(P) \tag{14}$$

$$\sigma_Y = \text{std}(Y) \tag{15}$$

$$\sigma_R = \text{std}(R) \tag{16}$$

The average of these standard deviations, denoted $\overline{\sigma_{\text{angle}}}$, is then computed as:

$$\overline{\sigma_{\text{angle}}} = \frac{\sigma_P + \sigma_Y + \sigma_R}{3} \tag{17}$$

Next, the variability of angular velocities is determined. First-order differences, approximating angular velocities, are computed for each angle sequence (for $N > 1$):

$$\Delta P_j = \theta_{p,j} - \theta_{p,j-1} \quad \text{for } j = 2, \ldots, N \tag{18}$$

and similarly for $\Delta Y_j$ and $\Delta R_j$. Let $\Delta P, \Delta Y, \Delta R$ be these sequences of differences. The variance of these differences is then calculated:

$$V_{\Delta P} = \text{var}(\Delta P) \tag{19}$$

$$V_{\Delta Y} = \text{var}(\Delta Y) \tag{20}$$

$$V_{\Delta R} = \text{var}(\Delta R) \tag{21}$$

The average variance of these angular differences, $\overline{V_{\Delta\text{angle}}}$, is given by:

$$\overline{V_{\Delta\text{angle}}} = \frac{V_{\Delta P} + V_{\Delta Y} + V_{\Delta R}}{3} \tag{22}$$

The variability of the head's position is assessed using the translation coordinates. Their variances are computed:

$$V_{T_x} = \text{var}(T_x) \tag{23}$$

$$V_{T_y} = \text{var}(T_y) \tag{24}$$

The average variance of translations, $\overline{V_{\text{trans}}}$, is then:

$$\overline{V_{\text{trans}}} = \frac{V_{T_x} + V_{T_y}}{2} \tag{25}$$

Finally, the Head Motion Dynamics score is computed by combining these components:

$$\textit{Head Motion Dynamics} = \sqrt{(\overline{\sigma_{\text{angle}}} \cdot \overline{V_{\Delta\text{angle}}}) + \overline{V_{\text{trans}}}} \tag{26}$$

**(6) Eyebrow Dynamics**   This metric measures the variability of eyebrow movements. For each frame $j$, facial landmarks for the eyebrows and eyes are detected. A representative vertical distance $d_{eb,j}$ between the eyebrows and the eyes is calculated. This distance is normalized by the inter-ocular distance $d_{io,j}$ (distance between the centers of the eyes) in that frame to account for face scale and head distance from the camera:

$$d'_{eb,j} = \frac{d_{eb,j}}{d_{io,j}} \tag{27}$$

The Eyebrow Dynamics score is the standard deviation of this normalized relative distance over all $N$ frames:

$$\textit{Eyebrow Dynamics} = \sqrt{\frac{1}{N-1} \sum_{j=1}^{N} (d'_{eb,j} - \overline{d'_{eb}})^2} \tag{28}$$

where $\overline{d'_{eb}}$ is the mean of $d'_{eb,j}$ over all frames.

### B.3 SYNCHRONIZATION METRICS

**(7) Silent Lip Stability**   This metric evaluates the stability of mouth closure during sustained silent periods, using a robust estimator to reduce the impact of outlier frames. First, the audio is analyzed using the Silero-VAD model to identify silent segments. Only segments with a duration of at least 300 ms are retained. Let $S_{\text{silent}}$ denote the set of frame indices belonging to these segments.

For each frame $j \in S_{\text{silent}}$, facial landmarks are detected using MediaPipe Face Mesh. For $P$ predefined pairs of upper and lower lip landmarks, the vertical distances are calculated, normalized by the inter-ocular distance $d_{io,j}$ to account for face scale. The per-frame average mouth opening is then:

$$d_{lip,j} = \frac{1}{P} \sum_{p=1}^{P} \frac{|y_{\text{upper},p,j} - y_{\text{lower},p,j}|}{d_{io,j}} \tag{29}$$

where $y_{\text{upper},p,j}$ and $y_{\text{lower},p,j}$ are the vertical coordinates of the $p$-th upper and lower lip landmarks in frame $j$.

We then employ the Median Absolute Deviation (MAD) is then employed to measure the variability of the lips movements during the silence period:

$$\text{MAD}_{lip} = \text{median}(|d_{lip,j} - \tilde{d}_{lip}|), \qquad j \in S_{\text{silent}} \tag{30}$$

where $\tilde{d}_{lip}$ is the median of $d_{lip,j}$ over all $j \in S_{\text{silent}}$.

The Silent Lip Stability score is given by this MAD value:

$$\textit{Silent Lip Stability (Robust)} = \text{MAD}_{lip} \tag{31}$$

**(8) Lip-Sync** The Lip-Sync metric quantifies the temporal alignment between mouth movements and the speech signal. First, the audio is extracted from the video, and Voice Activity Detection (VAD) is performed using the Silero-VAD model to identify speech segments. Let $S_{\text{speech}}$ denote the set of frame indices corresponding to detected speech frames.

For each frame $j$, the mouth openness $o_j$ is computed using facial landmarks from MediaPipe Face Mesh. The vertical distance between the mean position of the upper lip landmarks and the mean position of the lower lip landmarks is normalized by the inter-ocular distance $d_{io,j}$ to account for face scale:

$$o_j = \frac{\left|\bar{y}_{\text{upper},j} - \bar{y}_{\text{lower},j}\right|}{d_{io,j}} \tag{32}$$

Simultaneously, the short-time root-mean-square (RMS) energy of the audio signal is computed to obtain a per-frame speech volume $v_j$. Both $o_j$ and $v_j$ are min-max normalized over $S_{\text{speech}}$:

$$o_j^* = \frac{o_j - \min(o_{S_{\text{speech}}})}{\max(o_{S_{\text{speech}}}) - \min(o_{S_{\text{speech}}}) + \epsilon}, \qquad v_j^* = \frac{v_j - \min(v_{S_{\text{speech}}})}{\max(v_{S_{\text{speech}}}) - \min(v_{S_{\text{speech}}}) + \epsilon} \tag{33}$$

where $\epsilon$ is a small constant added to avoid division by zero.

The Lip-Sync score is the mean absolute difference between the normalized mouth openness and normalized speech volume over all frames in $S_{\text{speech}}$:

$$Lip\text{-}Sync = \frac{1}{|S_{\text{speech}}|} \sum_{j \in S_{\text{speech}}} \left| o_j^* - v_j^* \right| \tag{34}$$

## C APPENDIX: ADDITIONAL EXPERIMENTS ON SYNCNET INSTABILITY

Our experiments showed that Syncnet LSE-C and LSE-D can be influenced by the way audio and video are encoded. Indeed, when changing the audio encoding from **mp4a** to **mpga**, the LSE-D and LSE-C vary significantly. When tested on the entire HDTF dataset, we notice that the average absolute difference in LSE-D and LSE-C between videos with **mp4a** or **mpga** audio is 0.4. This absolute difference can even reach values as high as 1.2 for some samples. We observe similar results when comparing video using **H.264** and **H.265** encodings. In both experiments there are no noticeable qualitative differences from a human evaluation standpoint. This confirms the findings of (Yaman et al., 2024b) that Syncnet is not stable and can be influenced by various factor unrelated to lips synchronization.

## D TEMPORAL DRIFT IN OMNIAVATAR

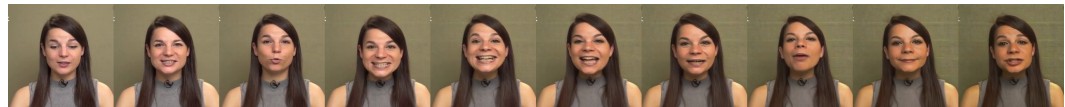

Figure 6: Temporal drift in OmniAvatar outputs over time. Ten frames are shown, sampled every 75 frames, illustrating gradual facial identity divergence, increasing visual artifacts, and the emergence of a color cast in longer videos.

## E APPENDIX: DESCRIPTION OF STATE-OF-THE-ART TH-GENERATION METHODS USED FOR BENCHMARKING

**Controltalk** (Zhao et al., 2024) is a talking face generation method to control face expression deformation based on driven audio, constructing the head pose and facial expression (lip motion) for both single image or sequential video inputs in a unified manner.

**DaGAN** (Hong et al., 2022) is a Depth-aware Generative Adversarial Network that first recovers dense 3D facial geometry (i.e., depth maps) from face videos in a self-supervised manner. This depth information is then used to guide the estimation of sparse facial keypoints and to learn a 3D-aware cross-modal attention mechanism, improving the generation of accurate face structures and motion fields.

**Dimitra** (Chopin et al., 2025) is a diffusion based framework for TH generation that uses 3D motions as an intermediate step. It leverages audio features, phonemes and text to generate fully animated, realistic TH videos.

**EchoMimic** (Chen et al., 2024c) uses audio speech to drive landmark sequences and employs a Latent Diffusion Model to convert input images into an efficient latent representation that is driven with the landmark sequence. It generates realistic results at high resolution.

**EMOPortraits** (Drobyshev et al., 2024) builds upon the MegaPortraits model to enhance its capability for rendering intense and asymmetric facial expressions. It introduces architectural changes and a new training pipeline, including a novel dataset with extreme emotions (FEED), and incorporates a speech-driven mode, making it a multimodal framework for high-fidelity avatar animation.

**Hallo2** (Cui et al., 2024) generates long-duration, high-resolution audio-driven portrait animations. It uses a patch-drop technique for temporal consistency, vector quantization for high resolution, and supports textual prompts for expression control.

**LIA** (Wang et al., 2022b) is a self-supervised autoencoder that animates images by linear navigation in its latent space, removing the need for explicit structure representation. Motion is constructed by the linear displacement of latent codes, using a learned set of orthogonal motion directions.

**LIA-X** (Wang et al., 2025) is an interpretable portrait animator designed as an autoencoder that models motion transfer as a linear navigation of motion codes. It incorporates a Sparse Motion Dictionary to disentangle facial dynamics into interpretable factors, enabling a controllable 'edit-warp-render' strategy for precise manipulation of facial semantics.

**Liveportrait** (Guo et al., 2024) is an efficient video-driven portrait animation framework using an implicit-keypoint-based approach for good generalization and controllability. It features stitching and retargeting modules for precise control over elements including eye and lip movements with minimal computational cost.

**MCNet** (Hong & Xu, 2023) proposes a Memory Compensation Network to address ambiguities from dramatic motions in talking head generation. It learns a global facial meta-memory bank that provides structure and appearance priors. An implicit identity representation, learned from keypoints and features of the source image, is used to query this memory bank and compensate for warped features, particularly in occluded regions.

**FOM** (Siarohin et al., 2019) proposes novel motion representations for animating articulated objects by identifying and tracking object parts as regions rather than keypoints. In a fully unsupervised manner, it infers motion from the principal axes of these regions, disentangles shape and pose to prevent identity leakage, and models global background motion separately with an affine transformation.

**OmniAvatar** (Gan et al., 2025) is an audio-driven video generation model focused on creating full-body animations with adaptive and natural movements. It employs a LoRA-based training approach on a foundation model and introduces a multi-hierarchical, pixel-wise audio embedding strategy to enhance lip-sync accuracy and ensure audio features guide the entire body motion, not just the face.

**Real3DPortrait** (Ye et al., 2024) is a framework for realistic 3D talking portrait synthesis. It improves 3D reconstruction by distilling knowledge from a 3D face generative model into an image-to-plane network. It facilitates animation with a motion adapter and synthesizes a complete,

realistic video by individually modeling the head, torso, and background, supporting both audio and video-driven inputs.

**SadTalker** (Zhang et al., 2023) is a method for generating TH that produces realistic 3D motion coefficients for animated, audio-driven TH from a single image. It leverages full-image animation capabilities and utilizes pre-trained models to enhance the expressiveness and authenticity of the animated TH.

**Wav2Lip** (Prajwal et al., 2020) is a lip synchronization model for videos, aligning lip movements with audio segments for different identities in various settings. It uses a lip-sync discriminator based on Syncnet to enhance the precision of lip movements in TH videos. Wav2Lips does not generate the entire TH but only the mouth region. The generated mouth region is then integrated into the original video without altering the rest of the content.

**X-Portrait** (Xie et al., 2024) is a conditional diffusion model for expressive portrait animation. It uses a pre-trained Stable Diffusion model as a rendering backbone and achieves fine-grained motion control via ControlNet. It interprets dynamics directly from the raw driving video (implicit control) rather than relying on intermediate representations such as landmarks, and uses a cross-identity training scheme to mitigate identity leakage from the driver.

## F  APPENDIX: EXPLANATION OF EXISTING METRICS

In this section we elaborate on the existing metrics for video evaluation, as well as on their limitations.

**FID:** The Fréchet Inception Distance (FID) constitutes an improvement of the Inception Score (IS). is a metric designed to evaluate the quality of generated images or videos. FID is computed by first extracting features from real and generated images using an inception network. Then, the features are treated as samples from two multivariate Gaussian distributions (real and generated) and the Fréchet distance between the two distributions is computed.

The Fréchet distance measures the distance between a generated image set and a source dataset, and is calculated as

$$\text{FID} = ||\mu_r - \mu_g||_2^2 + \text{Tr}(\Sigma_r + \Sigma_g - 2(\Sigma_r \Sigma_g)^{1/2}), \tag{35}$$

where $\mu_r$ and $\mu_g$ are the mean feature vectors of the real and generated images respectively, $\Sigma_r$ and $\Sigma_g$ are the covariance matrices of the real and generated images respectively.

FID is highly dependent on the performance of inception network and assumes that the images features follow a Gaussian distribution which might not be true. FID is also biased when evaluated on a finite set due to the assumption of Gaussian distribution (Chong & Forsyth, 2020). To be accurate, FID must be evaluated on a set that is large enough which might not be possible for all generation tasks. When used for video evaluation FID will only evaluate independent frame quality without regards for the temporal coherency.

**FVD:** The Fréchet Video Distance (FVD) is similar to FID but uses a network adapted for videos to extract the features. FVD is calculated as

$$\text{FVD} = ||\mu_r - \mu_g||_2^2 + \text{Tr}(\Sigma_r + \Sigma_g - 2(\Sigma_r \Sigma_g)^{1/2}, \tag{36}$$

where $\mu_r$ and $\mu_g$ are the mean feature vectors of the real and generated videos respectively, $\Sigma_r$ and $\Sigma_g$ are the covariance matrices of the real and generated videos respectively.

FVD has the same limitations as FID, and despite using an adapted network FVD still tends to focus more on single frame quality than on temporal coherence which is essential to evaluate videos.

**IS:** The Inception Distance (IS), uses an inception network that gives the probability of an image to belong to a certain class. Then, it uses the Kullback-Leibler divergence to compute a score related to the quality and diversity of the generated images. Specifically, the score is calculated to evaluate two factor: Intra-Class Similarity (high-quality images should have a strong probability of belonging to a single class) and inter-Class Diversity (generated images should belong to a variety of classes) and is calculated as

$$\text{IS} = \exp\left(\mathbb{E}_{x \sim p_g}\left[\text{KL}(p(y|x)||p(y))\right]\right), \tag{37}$$

Where $x \sim p_g$ denotes that $x$ is sampled from the generated images distribution $p_g$, $p(y|x)$ is the conditional probability distribution over the classes given the image $x$ and $p(y)$ is the marginal probability distribution over the classes, computed as $\mathbb{E}_{x \sim p_g}[p(y|x)]$.

However similarly to FID, this metric is very reliant on the inception network. It is unable to evaluate the intra-class diversity and will not work with images of classes not seen during the training of the inception network.

**LMD-F:** The Landmark Distance Face (LMD-F) is a metric to evaluate TH videos. LMD-F computes the average euclidean distance between the facial landmarks extracted for a real videos and those of a generated one for the same conditioning input (e.g driving audio speech). For LMD-F all of the face landmarks are used. LMD-F is calculated as

$$\text{LMD-F} = ||x_f - x'_f||_2 \tag{38}$$

where $x_f$ and $x'_f$ are the facial landmarks of the real and generated video respectively.

While LMD-F has been shown to correlate better than other metrics with human evaluation (Zhang et al., 2024), it still suffers from being a direct comparison to the ground truth. Indeed, different head motions and expressions between the generated sequence and the ground truth will be strongly penalized. At the same time, this difference is expected since head motion and expression are only loosely correlated to the audio sequence. However, as long as both look natural, human evaluators will give a high rating to the video even if it is different from the ground truth.

**LMD-M:** The Landmark Distance Mouth (LMD-M) is a metrics to evaluate TH videos. LMD-M compute the average euclidean distance between the facial landmarks extracted for a real videos and those of a generated one for the same conditioning input (e.g driving audio speech). For LMD-M only the landmarks pertaining to the mouth area landmarks are used.

$$\text{LMD-M} = ||x_m - x'_m||_2 \tag{39}$$

where $x_m$ and $x'_m$ are the mouth landmarks of the real and generated video respectively.

While LMD-M has been shown to correlate better than other metrics with human evaluation (Zhang et al., 2024), it still suffers from being a direct comparison to the ground truth. This direct comparison causes small temporal lags to be penalized when it wouldn't be noticed by human evaluators according to the recommendation by the International Telecommunication Union (199, 1998).

**LPIPS:** The Learned Perceptual Image Patch Similarity (LPIPS) measures the perceptual similarity between two images and try to provide a score that align with human perception. LPIPS uses a pre-trained CNN to obtain deep-features and computes the similarity between these features. The LPIPS value of CNN layer $l$ is calculated as

$$\text{LPIPS}_l(x, x') = \sum_l w_l \cdot ||f_l(x) - f_l(x')||_2, \tag{40}$$

where $f_l(x)$ and $f_l(x')$ are the feature representations of the real image $x$ and the generated image $x$ at layer $l$, $w_l$ are the weights of layer $l$. The final LPIPS score is a weighted sum of the $\text{LPIPS}_l$ across all the layers of the network.

While LPIPS aligns better with human evaluation it is still very dependent on the pre-trained network and is sensitive to image alignment.

**PSNR:** The Peak Signal-to-Noise Ratio (PSNR) compares two images at the pixel level by measuring the ratio between the maximum possible power of a signal (the original image) and the power of corrupting noise (the generated image). It is calculated as

$$\text{PSNR} = 10 \cdot \log_{10} \left( \frac{\text{MAX}^2}{\text{MSE}} \right) \tag{41}$$

$$\text{MSE} = \frac{1}{mn} \sum_{i=1}^{m} \sum_{j=1}^{n} [I(i,j) - K(i,j)]^2, \tag{42}$$

where MAX is the maximum possible pixel value of the image (usually 255), $I(i,j)$ represents the pixel value at position $(i,j)$ in the original image, and $K(i,j)$ represents the pixel value at position $(i,j)$ in the reconstructed image. The sums are taken over all pixels in the $m \times n$ image.

PSNR does not take the structure of the image into account, is very sensitive to noise and to outliers which can lead to low correlation with human evaluation.

**SSIM:** The Structural Similarity (SSIM) is a score that evaluates the similarity between two images. It is obtained by combining three components : the difference in brightness between the images, the difference in contrast between the images and the structural similarities between the images across small patches. SSIM is calculated as

$$\text{SSIM}(x,y) = l(x,y) \cdot c(x,y) \cdot s(x,y) \tag{43}$$

$$l(x,y) = \frac{2\mu_x\mu_y + C_1}{\mu_x^2 + \mu_y^2 + C_1} \tag{44}$$

$$c(x,y) = \frac{2\sigma_x\sigma_y + C_2}{\sigma_x^2 + \sigma_y^2 + C_2} \tag{45}$$

$$s(x,y) = \frac{\sigma_{xy} + C_3}{\sigma_x\sigma_y + C_3}, \tag{46}$$

where $\mu_x$ and $\mu_y$ are the means of the images $x$ and $y$, $\sigma_x$ and $\sigma_y$ are the standard deviations of the images $x$ and $y$, $\sigma_{xy}$ is the covariance between the images $x$ and $y$, $C_1$, $C_2$, and $C_3$ are small constants to stabilize the division when the denominators are close to zero. $l(x,y)$, $c(x,y)$ and $s(x,y)$ correspond to the luminance comparison, contrast comparison and structure comparison respectively.

SSIM need the images to be perfectly aligned in order to be accurate. Also, since it use small patches, it focus on local structure rather than global which lead to low correlation with human perception on complex images.

**Syncnet:** Syncnet (Chung & Zisserman, 2016) is a CNN-based network, aims to capture the correlation between audio and spatio-temporal features of the mouth region, calculating the audio offset (the number of frames by which audio is early or late compared to video). Its distance (LSE-D) and confidence score (LSE-C) are widely use to evaluate audio-lip synchronization in TH video. While Syncnet is good at evaluating the audio offset, finding the speaker in a video containing multiple persons or detecting unrelated audio (e.g dubbing) it is less useful when comparing two videos with similar lip synchronization (e.g videos generated by two different methods). In fact it has been shown that LSE-C and LSE-D have very limited correlation with human evaluation (Zhang et al., 2024). Some recent methods (Xu et al., 2024a) were even able to outperform the ground truth by a large margin on these metrics, showcasing their limitations. Additionally recent works (Yaman et al., 2024b;a) have shown that Syncnet is not stable and can easily be influenced by factors outside of lip synchronization (e.g mouth cropping, image quality, brightness...) making it difficult to apply on the diverse datasets used today. Additionally, our own experiments have shown that Syncnet is sensitive to audio and video encoding even when there are not noticeable difference for a human observer (Appendix C)

**LSE-D:** The Syncnet Distance (LSE-D) compute the distance between audio and video features at the offset predicted by Syncnet. See **Syncnet** entry for limitations.

**LSE-C:** The Syncnet Confidence score (LSE-C) computes the difference between the minimum and the median of the features distances over all possible offsets ($-10 \leq offset \leq 10, \quad offset \in \mathbb{Z}$). See **Syncnet** entry for limitations.

# G  USE OF LARGE LANGUAGE MODELS

We clarify the involvement of large language models (LLMs) is only for improving and polishing the manuscript

