# OpenReview forum: "THEval. Evaluation Framework for Talking Head Video Generation"
_ICLR.cc/2026/Conference — ICLR 2026 Conference Withdrawn Submission_

### Official Review · Reviewer_EBES · 2025-10-17

**Soundness:** 3
**Presentation:** 3
**Contribution:** 2
**Rating:** 2
**Confidence:** 3

**Summary:**

The authors define eight dimensions to evaluate talking head generation methods. They test 17 talking head generation methods and report the scores using the eight dimensions. They collect 5,011 videos from YouTube to evaluate the methods.

**Strengths:**

1. They conduct a large-scale study of 17 talking head methods and evaluate them thoroughly.
2. Talking head generation is an active research area, so the idea for a holistic evaluation suite makes sense.

**Weaknesses:**

1. Global Aesthetic Score is the same as the CLIP Aesthetic score with a different model[1].
2. There are missing details about human evaluation and assumptions about metrics, which I am doubtful about.


[1] CLIP knows image aesthetics: Simon Hentschel, Konstantin Kobs, Andreas Hotho.

**Questions:**

1. In Lip Dynamics, how is M selected?
2. Raw Euclidean distances change with face scale, camera zoom, etc. Without normalization, differences between frames will reflect geometry changes, not true lip motion. While it has been done for other metrics, its not clear if this was taken into account for Lip dynamics metric.
3. I think in the Eyebrow Dynamics metric, the authors equate variability with quality. A static eyebrow, like a neutral face, could be high-quality video yet score low, but an over-animated eyebrow could score high despite being unrealistic.
4. From my understanding, it seems the core idea of the Lip-Sync metric is that mouth openness should be proportional to audio volume. Why would it be so?
5. The phrase "all combinations of the seventeen state-of-the-art methods" is ambiguous and practically difficult to achieve. The details are missing as to how many samples were evaluated by humans, whether they were sampled randomly, or whether they evaluated all combinations. How many samples did a single human evaluate? Metrics like intern-annotator reliability should be used to evaluate the reliability of the human evaluations. The instruction "Please watch both videos and select which one looks more realistic" is intentionally simple to "minimize cognitive load." However, "realism" is a very broad term and could be interpreted differently by different users. It would be helpful if the authors could clear up these details.
6. How do the authors ensure the YouTube videos are not seen by any of these methods? To my knowledge, there exist some works that collect YouTube videos for training Talking head models[1,2]


[1] Flow-guided One-shot Talking Face Generation with a High-resolution Audio-visual Dataset: Zhimeng Zhang, Lincheng Li, Yu Ding, Changjie Fan

[2] Towards Generating Ultra-High Resolution Talking-Face Videos with Lip synchronization: Anchit Gupta, Rudrabha Mukhopadhyay, Sindhu Balachandra, Faizan Farooq Khan, Vinay P. Namboodiri, C. V. Jawahar

**Details Of Ethics Concerns:**

It is unclear whether the YouTube videos downloaded had an appropriate license.

---

### Official Review · Reviewer_Mg5X · 2025-10-23

**Soundness:** 2
**Presentation:** 3
**Contribution:** 2
**Rating:** 4
**Confidence:** 4

**Summary:**

In the talking head generation task, to address the issue that metric methods cannot accurately measure model performance, the authors propose the THEval benchmark. This benchmark includes 5k videos and proposes multiple metrics from three perspectives: quality, naturalness, and synchronization. The authors tested many state-of-the-art talking head generation methods and measured the gap between the benchmark and human preferences.

**Strengths:**

- The paper is well motivated and provides extensive experimental validations, covering many latest methods.
- The paper tests the correlation with human preferences, verifying that THEval metrics are more capable of reflecting model performance than existing metrics.

**Weaknesses:**

- The paper claims that THEval can "evaluate the improvement of generative methods". However, this benchmark only targets the driving task of speakers from the shoulders up. Currently, the development of talking head generation (THG) is gradually moving towards full-body driving and even multi-person driving, so this benchmark may be difficult to fully evaluate the progress in this field.
- The content of THG largely depends on the reference image. If the reference image itself has poor aesthetic quality, it may cause the metric to fail. In addition, the metric also struggles to reflect identity preservation ability.
- The evaluation methods for both Head Motion and Eyebrow Dynamics are somewhat unconvincing. Using global averages will erase detailed movements. The rationality of such evaluation requires more experimental support.
- These metrics should not only align with the coarse-grained human preference of "which is better" but should also truly reflect what they are intended to measure, thus helping people identify where the flaws of a solution lie.

**Questions:**

LSE-C and human preference show a negative correlation, which is somewhat counterintuitive. Please provide some examples where LSE-C is high but the Lip-Sync metric is low.

---

### Official Review · Reviewer_ikAd · 2025-10-29

**Soundness:** 2
**Presentation:** 2
**Contribution:** 1
**Rating:** 2
**Confidence:** 3

**Summary:**

The paper proposes an evaluation framework for the talking head video generation task by combining 8 metrics focusing on quality, naturalness, and synchronization. Moreover, the paper releases a benchmark for talking head evaluation. Experimental results show an improved correlation between the proposed evaluation score and human ratings compared with previous metrics.

**Strengths:**

1. The proposed idea is clearly illustrated and easy to follow.
2. Experimental results show improved correlation between the proposed metric and human ratings, and the proposed benchmark is evaluated on various talking head models.

**Weaknesses:**

1. I have several concerns about the proposed metric framework:
- One of the major challenges in talking head video generation is aligning mouth motion with audio, but there is a lack of metric for this.
- Facial feature details are not considered in the framework, such as emotional expression and alignment among the eyes, mouth, and head movements. These details could be captured using facial landmarks or intermediate feature maps from the model, which are not included in the proposed framework.

2. For the proposed THEVAL benchmark:
- The discussion is limited and lacks analysis. For example, what is the distribution of head motion, language, user identity, lip motion, etc.?
- There is also a lack of discussion regarding experiments on the THEVAL benchmark. For instance, what are the differences between the proposed and existing benchmarks? How does the THEVAL benchmark align with the proposed metric framework? What findings are observed from the experimental results?

**Questions:**

Questions for the proposed metric framework:
1. How are the 8 metrics combined? The 3 dimensions should have different significance in representing human preference—are any weights assigned to the different metrics?
2. For the human rating correlation experiment, what model is used to generate the videos? And do the videos adequately cover the three dimensions considered in the proposed metric framework?

**Details Of Ethics Concerns:**

Videos of public YouTube channels are collected for the evaluation dataset, which might have ethical issues for privacy.

---

### Official Review · Reviewer_aH21 · 2025-11-01

**Soundness:** 3
**Presentation:** 2
**Contribution:** 3
**Rating:** 4
**Confidence:** 2

**Summary:**

The paper introduces THEval, a comprehensive evaluation framework for talking head video generation. Unlike existing metrics that focus narrowly on image quality or lip synchronization and show poor correlation with human judgment, THEval combines eight algorithmic and perceptual metrics across three dimensions: quality, naturalness, and synchronization. The framework is validated on a large benchmark dataset and demonstrates a high correlation with human ratings, making it a more reliable and interpretable tool for assessing the performance of state-of-the-art audio- and video-driven talking head models.

**Strengths:**

1. The proposed THEval framework is highly useful for the evaluation of talking head video generation, as it integrates eight fine-grained metrics covering quality, naturalness, and synchronization, and demonstrates strong alignment with human perceptual judgments.

2. The experimental section is thorough, benchmarking 17 state-of-the-art models on a large, diverse dataset of 85,000 videos, and includes comprehensive user studies to validate the effectiveness of THEval against existing metrics.

3. The paper is well-structured and clearly written, with a logical flow from motivation and related work to the detailed presentation of the framework, followed by extensive experiments and analysis, making the contributions easy to understand and follow.

**Weaknesses:**

1. The presentation in the paper is somewhat insufficient. Although multiple metrics are proposed, there is no corresponding case study to intuitively demonstrate how each metric evaluates the generated videos.
2. The comparison with previous work does not seem to be comprehensive. In the field of talking head generation, there are some commonly used metrics, such as synchronization-related metrics (Sync-C and Sync-D) and quality-related metrics (introduced by Q-align). These metrics are more closely related to some of the metrics proposed in the paper, but they are not reflected in Table 2.

**Questions:**

1. I am somewhat confused about the design principles of the NATURALNESS-related metrics. For example, how was the formula for metric (3) derived? Additionally, since the authors use the magnitude of motion to measure naturalness, does this mean that more dramatic changes will result in higher scores?

2. Regarding the calculation of the final score, is it simply the average of all normalized metric scores? In the field of talking head generation, I believe lip quality and readability are the most important aspects. Should these metrics be assigned higher weights? Is there a risk that a model with high naturalness but poor readability could achieve a high final score?

---

### Note · Authors · 2025-11-13

I have read and agree with the venue's withdrawal policy on behalf of myself and my co-authors.